

# Morphometric properties of alternate bars and water discharge: a laboratory investigation

Marco Redolfi[1], Matilde Welber[1], Mattia Carlin[1], Marco Tubino[1], and Walter Bertoldi[1]

[1]Department of Civil, Environmental and Mechanical Engineering, University of Trento, Italy

**Correspondence:** Marco Redolfi (marco.redolfi@unitn.it)

**Abstract.** The formation of alternate bars in straightened river reaches represents a fundamental process of river morphodynamics that has received great attention in the last decades. It is well-established that migrating alternate bars arise from an autogenic, instability mechanism occurring when the channel width-to-depth ratio is sufficiently large. While several empirical and theoretical relations for predicting how bar height and length depend on the key dimensionless parameters are available, there is a lack of direct, quantitative information about the dependence of bar properties on flow discharge. We performed a series of experiments in a long, mobile-bed flume with fixed and straight banks, at different discharges. The self-formed bed topography was surveyed, different metrics were analysed to obtain quantitative information about bar height and shape, and results were interpreted in the light of existing theoretical models. The analysis reveals that the shape of alternate bars highly depends on their formative discharge, with remarkable variations in the harmonic composition and a strong decreasing trend of the skewness of the bed elevation. Similarly, the height of alternate bars clearly decreases with the water discharge, in quantitative agreement with theoretical predictions. However, the disappearance of bars when discharge exceeds a critical threshold is not as sharp as expected, due to the formation of so-called "diagonal bars". This work provides basic information for modelling and interpreting short-term morphological variations during individual flood events and long-term trajectories due to alterations of the hydrological regime.

## 1 Introduction

Alternate bars are large-scale bedforms, characterized by a repetitive sequence of scour holes and depositional diagonal fronts with longitudinal spacing of the order of several channel widths, which are often observed in both sand and gravel bed rivers (e.g., Engels, 1914; Jaeggi, 1984; Rhoads and Welford, 1991; Church and Rice, 2009; Jaballah et al., 2015; Rodrigues et al., 2015).

They have been extensively studied in the last fifty years, because of both their practical and theoretical relevance. From the point of view of river engineering, alternate bars are often undesired for their erosional effect on banks and bridge piers and the depositional effect that can disturb navigation and increase flooding risk. From an ecosystem perspective, alternate bars represent one of the relevant morphological units, creating suitable habitats for aquatic fauna and riparian vegetation, largely contributing to habitat diversity (Gilvear et al., 2007; Zeng et al., 2015). Finally, from a theoretical point of view, they represent



a fascinating phenomenon, which plays a fundamental role in the dynamics of a variety of fluvial systems, such as meandering rivers, channel bifurcations and braided rivers (e.g., Lewin, 1976; Parker, 1976; Fredsoe, 1978).

A large number of studies (e.g., Hansen, 1967; Callander, 1969; Sukegawa, 1972; Parker, 1976; Fujita and Muramoto, 1982; Nelson, 1990) demonstrated that downstream-migrating alternate bars can spontaneously develop in a straight, channelized reach as the result of instability mechanisms of a cohesionless bed. Due to this autogenic formation mechanisms, this kind of

bedforms are often referred to as "free bars" (Seminara and Tubino, 1989).

Specifically, theoretical and laboratory experiments (Fredsoe, 1978; Jaeggi, 1984; Fujita and Muramoto, 1985; Colombini et al., 1987; Lanzoni, 2000a) identified the channel width-to-depth ratio as the key controlling parameter for the formation of free alternate bars: when the channel is relatively narrow, the effect of gravitational pull on the bed load transport is relatively strong and tends to suppress any transverse bed gradient; conversely, in relatively wide channels initially small, periodic

perturbations of the bed elevation generate a topographic steering of the flow field that in turn produces a growth of the bed perturbation itself, which leads to the spontaneous, self-sustained development of alternate bars. Therefore, it is possible to define a threshold value of the aspect ratio (i.e. the half width-to-depth ratio), $\beta_{cr}$, representing the lower limit for which alternate bars are expected to form.

However, when the width-to-depth ratio is smaller than the threshold value, the equilibrium bed configuration is not nec-

essarily planar, as other kind of bedforms may result from a different instability mechanisms, such as short, shallow and fast migrating three-dimensional bedforms, usually called "diagonal bars" (Einstein and Shen, 1964; Jaeggi, 1984; Colombini and Stocchino, 2012). Since the transition between alternate and diagonal bars is not always sharp, and since they are both characterized by a diagonal pattern, they can be easily confused. However, as highlighted by Colombini and Stocchino (2012), diagonal bars represent a clearly distinct kind of bedforms and should be more properly regarded as three-dimensional oblique dunes.

In fact, they respond to a different formation mechanisms (e.g., they can not be described by shallow-water two-dimensional models), and they depend on different controlling parameters (e.g., they are directly influenced by water depth and Froude number).

On the other hand, when the aspect ratio becomes very large, transition to more complex, wandering and braiding multithread channels is observed (e.g., Fredsoe, 1978; Eaton et al., 2010; Ahmari and Da Silva, 2011; Garcia Lugo et al., 2015),

which poses an upper limit to the range of $\beta$ values where free alternate bars are expected to form.

Under steady flow conditions, free bars can attain an equilibrium state, where they simply migrate downstream without changing their morphology (Ikeda, 1984; Colombini et al., 1987). Different theoretical and empirical relations for estimating the equilibrium bar height and wavelength are available (e.g., Ikeda, 1984). Specifically, weakly nonlinear theories (Colombini et al., 1987; Bertagni and Camporeale, 2018) allow for a physically based, analytical prediction of how the equilibrium bar

configuration depends on the dimensionless channel and flow parameters.

Nevertheless, there is basically no direct, quantitative analysis about how equilibrium properties of alternate bars depend on water discharge. In particular, very few data exist about the shape of alternate bars, as previous experiments manly focused on bar height, wavelength and growth rate (e.g., Ikeda, 1984; Jaeggi, 1984; Fujita and Muramoto, 1985; Lanzoni, 2000a). Moreover, there is little knowledge about the transition from alternate bars to plane-bed or diagonal bars configurations that





may occur when varying the flow discharge. This lack of information makes it difficult to understand how changes in the flow regime may alter the bed morphology.

In this work we follow an integrated experimental and theoretical approach, to address the following research questions: (i) how do geometrical properties of alternate bars depend on water discharge? (ii) is it possible to identify different bar styles depending on flow conditions? (iii) is there a sharp transition from alternate bar morphology to a plane-bed configuration? To

answer these questions we performed a series of flume experiments with identical channel conditions and sediment characteristics, but different flow discharge, and we compared experimental results with theoretical predictions from the weakly nonlinear model of Colombini et al. (1987).

## 2  Methods

### 2.1  Laboratory setup

Laboratory experiments were carried out in a $24\,\mathrm{m}$ long flume at the Hydraulics Lab of the University of Trento. The physical model consisted of a straight channel of width $W = 0.3\,\mathrm{m}$, with vertical banks built out of plywood covered by a thick plastic tarp. Uniform sand with a median diameter of $d_{50} = 1.01\,\mathrm{mm}$ was used as feed and bed material. Discharge and sediment input to the flume was set automatically using a recirculating pump and a calibrated screw feeder. At the downstream end of the flume, the output bed load accumulated in a large filtering crate resting on four load cells was weighted every 10 seconds.

A laser profiler moving on high-precision rails was used to map the topography of the drained bed with vertical accuracy of $0.1\,\mathrm{mm}$ and spatial resolution $50 \times 5\,\mathrm{mm}$ (longitudinal and transversal direction, respectively).

We performed a set of 16 steady flow runs with discharge ranging from $Q = 0.5$ to $4.2\,\mathrm{l\,s^{-1}}$; all but two of the discharge values ($1.5$ and $4.2\,\mathrm{l\,s^{-1}}$) were repeated twice to obtain a larger dataset of bed topographies. The chosen discharge values ensure a wide range of channel aspect ratios, and are associated with the hydraulic conditions reported in Table 1.

At the beginning of each model run, the bed was screeded at a slope $S = 0.01$ using a blade mounted on a movable trolley. Sediment supply for each run was first assigned on the basis of previous experiments carried out with a similar setup (Garcia Lugo et al., 2015) and gradually adjusted during the transient phase of the run to match bed load output. The duration of experimental runs was chosen to ensure equilibrium conditions. Specifically, following the criteria proposed by Garcia Lugo et al. (2015) the run duration was 10–20 times the Exner timescale.

Wetted width ($W_w$) and active width ($W_a$) were measured at 20 regularly spaced cross sections, twice per each run, and averaged in space and time. Migration rate of the alternate bars was estimated by tracking the position of up to 15 individual bar fronts at fixed time intervals. At the end of the run the bed was drained to acquire topography data. Laser surveys covered a $20.5\,\mathrm{m}$ long area starting $2\,\mathrm{m}$ downstream of the inlet to exclude the effect of local disturbances.

The average sediment flux, $Q_s$, was estimated on the basis of the total weight of the transported material, excluding the first

transitory part of the experiment (an equilibrium condition was considered achieved when the cumulative mean of the bed load signal got within $5\,\%$ of the global mean).





**Table 1.** Summary data of the laboratory experiments. Channel width, slope and median grain size are constant and equal to $W = 0.30\,\mathrm{m}$, $S = 1.0\,\%$ and $d_{50} = 1.01\,\mathrm{mm}$, respectively. Water depth, Froude number and aspect ratio are computed by assuming uniform flow conditions over a plane bed.

| Case # | 1 | 2 | 3 | 4 | 5 | 6 | 7 | 8 | 9 |
|---|---|---|---|---|---|---|---|---|---|
| Discharge $Q$ [$\mathrm{l\,s^{-1}}$] | 0.5 | 1.0 | 1.5 | 2.0 | 2.5 | 2.7 | 3.0 | 3.4 | 4.2 |
| Run duration $T$ [h] | 20 | 10 | 6 | 5 | 5 | 5 | 5 | 5 | 2 |
| Sediment transport $Q_s$ [$\mathrm{cm^3\,s^{-1}}$] | 0.08 | 0.45 | 1.22 | 1.81 | 2.47 | 2.59 | 3.04 | 3.63 | 4.97 |
| Relative wetted area $W_w/W$ [$-$] | 0.97 | 1.00 | 1.00 | 1.00 | 1.00 | 1.00 | 1.00 | 1.00 | 1.00 |
| Relative active area $W_a/W$ [$-$] | 0.39 | 0.76 | 0.88 | 0.95 | 0.98 | 1.00 | 1.00 | 1.00 | 1.00 |
| Water depth $D$ [cm] | 0.72 | 1.06 | 1.34 | 1.58 | 1.79 | 1.87 | 1.99 | 2.14 | 2.42 |
| Froude number $Fr$ [$-$] | 0.86 | 0.96 | 1.02 | 1.06 | 1.09 | 1.10 | 1.12 | 1.13 | 1.17 |
| Aspect ratio $\beta = W/(2D)$ [$-$] | 21.3 | 14.4 | 11.4 | 9.7 | 8.5 | 8.1 | 7.7 | 7.1 | 6.3 |
| Critical aspect ratio $\beta_{cr}$ [$-$] | 3.2 | 5.0 | 5.9 | 6.5 | 7.0 | 7.1 | 7.3 | 7.5 | 7.9 |
| Resonant aspect ratio $\beta_{res}$ [$-$] | 4.2 | 7.0 | 8.7 | 10.0 | 11.0 | 11.4 | 11.9 | 12.6 | 13.8 |

## 2.2 Topography data processing

Laser surveys were first processed by removing points falling outside the channel bed and by subtracting the average longitudinal slope. This allowed for obtaining DEMs of the detrended bed elevation.

The investigation of the geometric properties of alternate bars required the identification of individual bedforms. To this aim, we applied the widely accepted definition of bar wavelength as the length between two successive troughs (Eekhout et al., 2013) and developed an automatic procedure to map the position of troughs on DEMs. Elevation maps of individual bars were obtained by splitting DEMs at trough points. Very irregular bars and bedforms that were only partially within the study area were excluded from further calculations. Bar DEMs were normalized by subtracting the elevation mean.

Finally, to facilitate the comparison of the shape of individual bars, each bar DEM was resampled along a $64 \times 64$ grid using an inverse distance weighted routine. This also allowed for the definition of a characteristic bar shape, referred to as the ensemble bar, obtained by computing the average elevation of each grid point across all the bars formed at the same discharge.

## 2.3 Different metrics to characterize bar properties

Alternate bars are commonly described in terms of their wavelength, height and migration rate. Bar wavelength is the distance
between consecutive, corresponding points along the flow direction. Bar height is usually defined as the vertical distance between the bottom of the pool and the top of the bar surface, with several method and metrics proposed in the literature. Finally, for freely migrating bars, the migration rate is the speed at which the bar front moves downstream. However, the geometrical properties of bars are not limited to their height and wavelength, as more detailed information about geometrical shape of bedforms can be derived by analyzing their elevation distribution. The geometrical properties of alternate bars in a
river channel can be investigated at reach scale or at bedform scale. In the first case, information is derived from the spatial





distribution of bed elevation for the entire reach, while in the second case, the reach is divided into segments corresponding to individual bars. The latter approach requires the identification of the spatial limits of individual bedforms and therefore the results may be influenced by the procedure used to define these limits. However, the resulting dataset provides not only reach-averaged values of geometry parameters but also information on their variability.

### 2.3.1 Metrics for bar height and bed relief

The most intuitive and widely used way to define bar height is the difference between the maximum and the minimum elevation along a bar unit, where the bed elevation $\eta$ is usually computed after removing the mean bed slope. Though different symbols have been used in the literature, we refer to the Ikeda (1984) notation, namely:

$$H_{BM} = max(\eta) - min(\eta), \tag{1}$$

where $\eta$ indicates the (detrended) bed elevation.

A slightly different definition of bar height (e.g., Fujita and Muramoto, 1985) is based on computing the elevation difference along individual cross sections ($H_{Bsec}$) and then taking its maximum value ($H_B$):

$$H_{Bsec} = max_{sec}(\eta) - min_{sec}(\eta), \qquad H_B = max(H_{Bsec}), \tag{2}$$

where $max_{sec}$ and $min_{sec}$ denote the maximum and the minimum elevation along individual cross sections.

The above definitions have a clear physical meaning, as they directly represent the bar height, from the crest to the trough. However, being based on extreme elevation values, such metrics are sensitive to outliers and measurement errors. Therefore, it is sometimes convenient to estimate the topographic effect of alternate bars through different metrics, which measure the "relief" rather than the bar height.

Specifically, the bed relief can be defined through the standard deviation of the elevation distribution:

$$SD = std(\eta), \tag{3}$$

or, alternatively, through the Bed Relief Index (e.g., Hoey and Sutherland, 1991; Liébault et al., 2013), which is defined on a cross-sectional basis as follows:

$$BRI_{sec} = std_{sec}(\eta), \qquad BRI = mean(BRI_{sec}), \tag{4}$$

where $std_{sec}$ indicates the standard deviation calculated along individual cross sections.

All of these metrics are applied at bar scale, although it is possible to compute them for elevation maps containing more than one bar wavelength. It is important to note that, while $H_{BM}$ and $SD$ are based on the full 2-D distribution of elevation, $H_B$ and $BRI$ are based on the elevation along individual cross sections. Since the highest and lowest points of a bar do not necessarily occur along the same cross section, $H_B$ and $BRI$ are expected to provide a lower estimate of height if compared to $H_{BM}$ and $SD$, respectively.





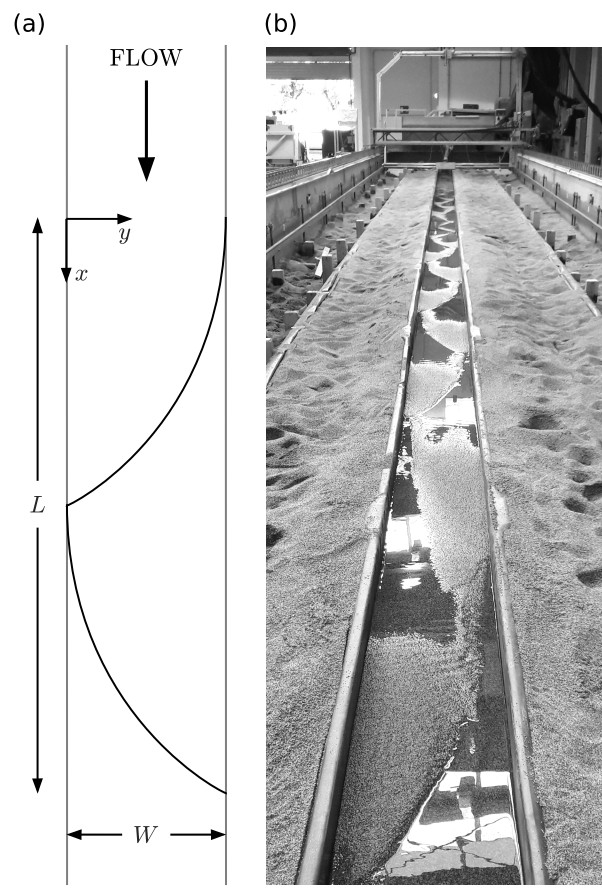

**Figure 1.** (a) reference system $(x, y)$ for and individual bar of wavelength $L$ in a straight channel of width $W$, with the curved line indicating the typical position of bar fronts; (b) picture of the flume at the end of the experiment with $Q = 2.5\,\mathrm{l\,s^{-1}}$, showing the presence of alternate bars. Flow is from top to bottom.

### 2.3.2 Metrics for the bar shape

The shape of bars can be characterized in terms of the skewness parameter $(SK)$, which measures the asymmetry of the bed elevation distribution, thus providing information on the relative proportion of high and low areas within a bar. River bed elevation maps often show negative skewness, with deep, narrow channels carved into large, higher elevation bars (e.g., Bertoldi et al., 2011; Garcia Lugo et al., 2015).

Being based on the relative frequency of the elevation values, the above metrics do not to provide information about the spatial arrangement of the bedforms. To obtain synthetic information about the spatial structure of bars, we analysed the bed elevation maps through the two-dimensional Fourier transform (e.g., Garcia and Nino, 1994; Zolezzi et al., 2005). As detailed





in Appendix A, the topography of an individual bar of wavelength $L$ (see Fig. 1) can be represented as follows:

$$\eta(x,y) = \sum_{m=0}^{\infty} \sum_{n=0}^{\infty} |A_{nm}| \cos(\pi m y/W) \cos\left(2\pi n x/L + \phi_{nm}\right), \tag{5}$$

where $x$ is the longitudinal coordinate, $y$ is the transverse coordinate (with origin at the right bank), while $|A_{nm}|$ and $\phi_{nm}$ represent amplitude and phase of each Fourier component. The amplitude of the main components provides information about possible symmetry properties, the relative importance of two-dimensional and three-dimensional topographic effects, and the deviation from the simple sinusoidal structure that arises from linear stability analyses (e.g., Fredsoe, 1978).

## 2.4 Application of the weakly nonlinear theory of Colombini et al. (1987)

The theory of Colombini et al. (1987) is based on a weakly nonlinear solution of the two-dimensional shallow water and Exner model for a straight channel of constant width and downstream gradient.

Application of the theory required to specify closure relations for flow resistance and sediment transport. First, the dimensionless Chézy coefficient, $c$, was expressed through the widely used logarithmic friction formula of Engelund and Fredsoe (1982):

$$c = 6 + 2.5 \log\left(\frac{D}{2.5\,d_{50}}\right), \tag{6}$$

where $D$ indicates the water depth. Second, the bed load transport rate per unit width, $q_b\,[\mathrm{m^2\,s^{-1}}]$, was quantified by means of the Parker (1978) formula:

$$q_b = 11.2\sqrt{g\Delta d_{50}^3}\,\theta^{1.5}\left(1 - \frac{\theta_{cr}}{\theta}\right)^{4.5}, \qquad \theta_{cr} = 0.03, \tag{7}$$

where $g$ is the gravitational acceleration, $\Delta = 1.65$ is the relative submerged weight of sediment, $\theta$ is the Shields number and
$\theta_{cr}$ indicates critical conditions for incipient sediment motion. This transport formula was chosen for two reasons: (i) it exhibits a critical threshold that is consistent with our experiments; (ii) it is suitable for analytical treatment, because for $\theta > \theta_{cr}$ it is continuous and has continuous derivatives.

Finally, the effect of the lateral bed slope on the direction of the bed load transport was modeled according to the Ikeda (1982) formulation:

$$\tan(\gamma) = -\frac{r}{\sqrt{\theta}}\frac{d\eta}{dy}, \tag{8}$$

where $\gamma$ is the angle between the velocity vector and the sediment transport vector, and $r$ is an empirical parameter. Calibration based on minimizing the difference between experimental and analytical values of $H_{BM}$ gave a value $r = 0.25$, which is only slightly lower than that proposed by Colombini et al. (1987) ($r = 0.3$).

Once closure relations, channel geometry and discharge are prescribed, the analytical model allows for the computation of
equilibrium bar topography and migration rate.



## 3   Results

The bed topographies obtained under different discharges are illustrated in Fig. 2. A regular pattern of bedforms can be recognized in all maps, with substantial differences in shape and relief.

At the lowest discharge ($0.5\,\mathrm{l\,s^{-1}}$) the bed shows a complex topography with alternate, elongated pools along the banks but few clearly discernible bar fronts and several small channels cutting the main bedforms. In these conditions the top of the bars ($3\,\%$ of the area, see Table 1) begins to emerge and less than half of the bed surface is actively transporting sediments.

At higher flows (1.0 and $1.5\,\mathrm{l\,s^{-1}}$), bed topography is dominated by a coherent pattern of bars with elongated pools and a sharp front that is almost transversal to the flow direction. Between $Q = 2.0$ and $2.7\,\mathrm{l\,s^{-1}}$, relief progressively decreases and bar fronts becomes curved and oblique in a regular fish-scale pattern. Finally, between $Q = 3.0$ and $4.2\,\mathrm{l\,s^{-1}}$, bars are shorter and shallower. It is also important to note that for increasing discharge, well defined bars progressively disappear from the upstream end of the channel, where the bed shows a superimposition of low-relief, irregularly spaced oblique fronts.

### 3.1   Bar height and bed relief

A comparison of metrics for bar height and bed relief is illustrated in Fig. 3a and 3b. The bar height $H_{BM}$ is maximum (almost $40\,\mathrm{mm}$) for the $Q = 1.0\,\mathrm{l\,s^{-1}}$ run, then it gradually decreases with discharge until it attains a relatively constant value of about $13\,\mathrm{mm}$ for $Q \geq 3.0\,\mathrm{l\,s^{-1}}$. At the lowest discharge ($0.5\,\mathrm{l\,s^{-1}}$), bar height is lower than the peak value, showing a value around $33\,\mathrm{mm}$. As expected, $H_B$ is smaller than $H_{BM}$ for all runs, but the difference is minimal and does not show a clear trend with discharge.

An analogous behavior is observed for the bed relief metrics $SD$ and $BRI$ (Fig. 3b). Specifically, $BRI$ tends to be only slightly smaller than $SD$, and both metrics exhibit a variation with discharge that follows the same trend observed for the bar height metrics $H_{BM}$ and $H_B$. However, the variation of $SD$ and $BRI$ with discharge is less gradual, with bars formed at $Q = 1.0$ and $1.5\,\mathrm{l\,s^{-1}}$ showing distinctively higher values (nearly $+50\,\%$) than the other cases. In general, values of $SD$ and $BRI$ are much lower than $H_{BM}$ and $H_B$, as bed relief metrics cover a range of values that is about one fifth the observed range of bar height.

Bars are downstream migrating with a speed of the order of a few millimeters per second. At the lowest discharge, migration rate was not measured because of the lack of easily recognizable fronts and the presence of complex patterns of erosion and deposition. For higher flows, migration rate rapidly increases with discharge from almost zero to $3\,\mathrm{mm\,s^{-1}}$ for $Q = 3.0\,\mathrm{l\,s^{-1}}$ (see Fig. 3c). Noteworthy, the two highest values of discharge exhibit a much higher migration rate of about $5\,\mathrm{mm\,s^{-1}}$.

Mean bar wavelength (see Fig. 3d) is higher at low flows (about 3 to $3.5\,\mathrm{m}$, corresponding to 10 to 12 channel widths) and decreases at higher discharges to approximately $1\,\mathrm{m}$ (about three channel widths). Specifically, the bar wavelength shows a rapid drop between $Q = 1.5$ and $2.0\,\mathrm{l\,s^{-1}}$, followed by a gradual decrease.



**Figure 2.** Maps of detrended bed elevation for increasing values of discharge. Flow is from top to bottom. Longitudinal scale is compressed for clarity.



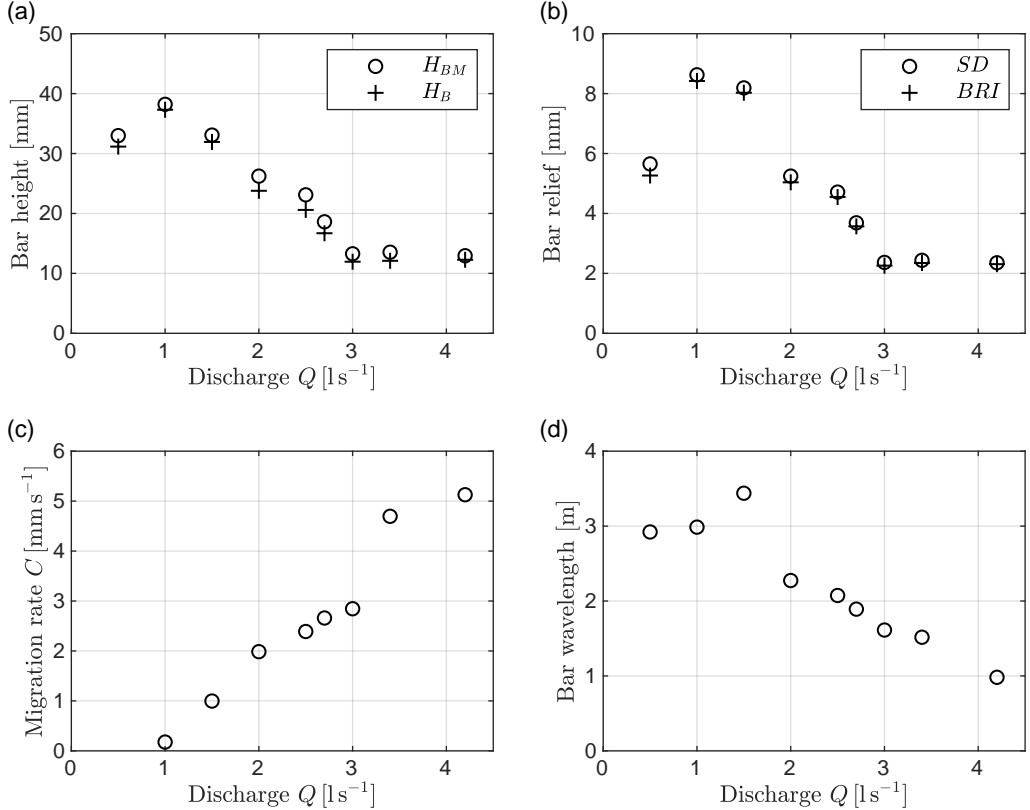

**Figure 3.** Mean properties of bars depending on discharge: (a) bar height $H_{BM}$ and $H_B$; (b) bed relief, as measured by the standard deviation of the bed elevation distribution ($SD$) and the bed relief index ($BRI$); (c) bar migration rate $C$; (d) bar wavelength $L$.

## 3.2 Predictions by the weakly nonlinear theory

The values of the equilibrium bar height predicted by the weakly nonlinear theory are reported in Fig. 4, which shows a decreasing trend of $H_{BM}$ with the water discharge, until it vanishes when the channel aspect ratio, $\beta$, matches its threshold value $\beta_{cr}$ (no bars). Therefore, it is possible to define a corresponding threshold value of the flow discharge (a "critical discharge",
$Q_{cr} = 3.171\,\mathrm{s}^{-1}$), which separates the formative conditions for alternate bars ($Q < Q_{cr}$) from the region where bars do not develop ($Q > Q_{cr}$).

We note that the weakly nonlinear theory is formally valid near the critical conditions, although the comparison with experimental data suggests its applicability within a wider range of conditions (Colombini et al., 1987; Lanzoni, 2000b). However, at relatively low values of discharge, the predicted equilibrium elevation of the top of the bars would exceed the water surface
elevation, which makes the equilibrium value of $H_{BM}$ no longer meaningful. Therefore, when the discharge is smaller than the so-called fully-wet threshold, $Q_{fw}$ (see Adami et al., 2016) the system cannot reach equilibrium bar height. Under these conditions it is then reasonable to assume that bar growth stops as bar tops start to emerge, and to set the maximum bar ele-





vation equal to the water surface elevation. This concept of emersion-limited bar height is represented by the dash-dot line in Fig. 4, which shows an opposite (i.e. increasing with the discharge) trend with respect to the theoretical equilibrium height.

As shown in Fig. 4 a third relevant threshold, i.e the flow discharge corresponding to incipient sediment motion, $Q_i = 0.26\,\mathrm{l\,s^{-1}}$, defines the lowest limit of the region of possible bar formation.

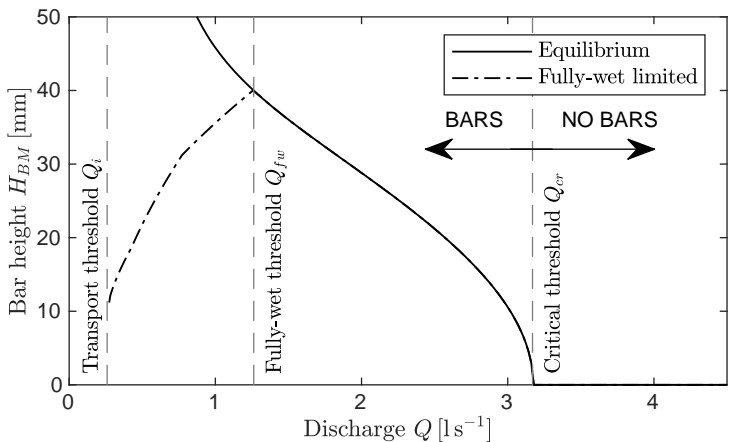

**Figure 4.** Bar height as a function of discharge, according to the weakly nonlinear theory of Colombini et al. (1987). Solid line represents the equilibrium solution, while the dash-dot line indicate the bar height we obtained by limiting the bar growth to the fully-wet condition. The development of alternate bars highly depends on discharge state with respect to the three key thresholds (vertical dashed lines), which represent: (i) the critical condition of incipient sediment motion, $Q_i = 0.26\,\mathrm{l\,s^{-1}}$; (ii) the conditions for which bars at equilibrium are fully wet, $Q_{fw} = 1.26\,\mathrm{l\,s^{-1}}$; and (iii) the critical condition for bar formation, $Q_{cr} = 3.17\,\mathrm{l\,s^{-1}}$.

The theoretical response of bar height to varying flow conditions is then compared with the laboratory data, which gives the results illustrated in Fig. 5. Consistently with the theoretical analysis, all the metrics are represented in dimensionless form, by scaling bar height and relief with the water depth $D_0$, the bar wavelength by the channel width $W$, and the migration rate by

the flow velocity $U_0$. Moreover, we define a dimensionless discharge as:

$$\Delta Q^* = \frac{Q - Q_{cr}}{Q_{cr} - Q_i},\tag{9}$$

so that values from $-1$ to $0$ cover the entire range of bar formation, from the threshold of incipient sediment transport $Q_i$ to the critical threshold $Q_{cr}$.

From this comparison it is apparent that bars observed at $3.4$ and $4.21\,\mathrm{s^{-1}}$ are anomalous, for a number of reasons: (i)

they occur outside the region of bar formation (i.e. at $Q > Q_{cr}$); (ii) they exhibit a much faster migration rate, and (iii) their wavelength is much shorter with respect to the typically observed values ($L = 5$–$12\,W$, see Tubino et al., 1999). This type of bedform closely resembles the "diagonal bars" described by Jaeggi (1984) as three-dimensional mesoforms characterized by a wavelength of around three times the channel width, limited relief, shallow pools and a symmetrical elevation distribution. These bedforms were observed at Froude numbers close to one and did not match the region of alternate bar formation.





Experimental observations by Jaeggi (1984) suggested that diagonal bars can be considered as intermediate bed forms asso-
ciated with the transition of dunes from two- to three-dimensional configurations. This idea was confirmed by the theoretical
work of Colombini and Stocchino (2012), which provided an interpretation of diagonal bars as three-dimensional oblique
dunes, distinct from alternate bars. In what follows we will therefore refer to bars observed at $\Delta Q^* > 0$ as diagonal bars,
reserving the term "alternate bars" to the remaining cases.

The analytical model reproduces remarkably well both the bar height and the bed relief of alternate bars (Figures 5a and
5b). However, when the discharge approaches the critical threshold $Q_i$ the weakly nonlinear model is no longer valid and the
solution for the equilibrium amplitude diverges. As pointed out before, when discharge is lower than the fully-wet threshold
$Q_{fw}$ the singularity of the analytical solution can be mitigated by considering the fully-wet limited bar height, which provides
a reasonable estimate of $H_{BM}$ and $SD$.

Similarly, the bar migration rate (Fig. 5c) is well reproduced by the analytical model, both for the overall trend and for the
absolute values, although the theory significantly overestimates the observed value at $Q = 1.01\,\mathrm{s}^{-1}$. Furthermore, the theory
properly predicts the wavelength of bars only for intermediate values of discharge (see Fig. 5c), while it does not capture the
overall decreasing trend, and therefore sharply underestimates the length of bars observed in the three runs with $Q \leq 1.5\,\mathrm{l\,s}^{-1}$.

### 3.3   Quantitative analysis of the bar shape

In order to filter out the relatively small differences of single bar units, we computed for each discharge value an ensemble
bar shape, defined as the average topography of all the bars formed under the same flow conditions. The resulting ensemble
topographies represented in Fig. 6 show a rather regular pattern. We then analysed the response of the bar shape to changing
discharge by computing the skewness and the Fourier components of the ensemble bars for each discharge value.

    Fig. 7 shows that the skewness is always negative, which indicates a left-tailed bed elevation distribution. For the lowest
discharge the skewness is around $-1.5$, which matches typical values observed for wandering and braided channels (see Garcia
Lugo et al., 2015). Highly negative values of the skewness can be associated with the presence of narrow, deep troughs and
wide, relatively flat bar crests that are clearly detectable in Fig. 6 for the ensemble bars at $Q \leq 1.5\,\mathrm{l\,s}^{-1}$. This morphological
characteristic becomes progressively less pronounced, as the ensemble bars corresponding to the range $Q = 2.5\text{--}3.0\,\mathrm{l\,s}^{-1}$ show
a comparable extension of scour and deposition areas and the presence of distinct diagonal fronts.

The observed trend of the skewness parameter is in qualitative agreement with the theory, which predicts an increase from
negative values at low flow to vanishing values (i.e. nearly symmetrical distribution) when approaching the critical condition
for bar formation (i.e. $\Delta Q^* \simeq 0$). However, the magnitude of the observed skewness is much larger than the theoretical esti-
mate, due to the limited capability of the theoretical model to fully represents the complex, highly nonlinear morphodynamic
processes (see Colombini et al., 1987).

The analysis of the Fourier spectral composition of bed topography provides the amplitude of each component along the
transverse and longitudinal direction. An example is shown in Fig. 8 for the ensemble bar of the $Q = 2.5\,\mathrm{l\,s}^{-1}$ run. The plot
shows the amplitude of the first 36 ($6 \times 6$) harmonic components, identified by their longitudinal ($n$) and transverse ($m$) mode.
In this representation, $n = 1$ indicates a complete sinusoidal period in one bar wavelength, while $m = 1$ indicates half a wave



Earth **Surface**
**Dynamics**
Discussions

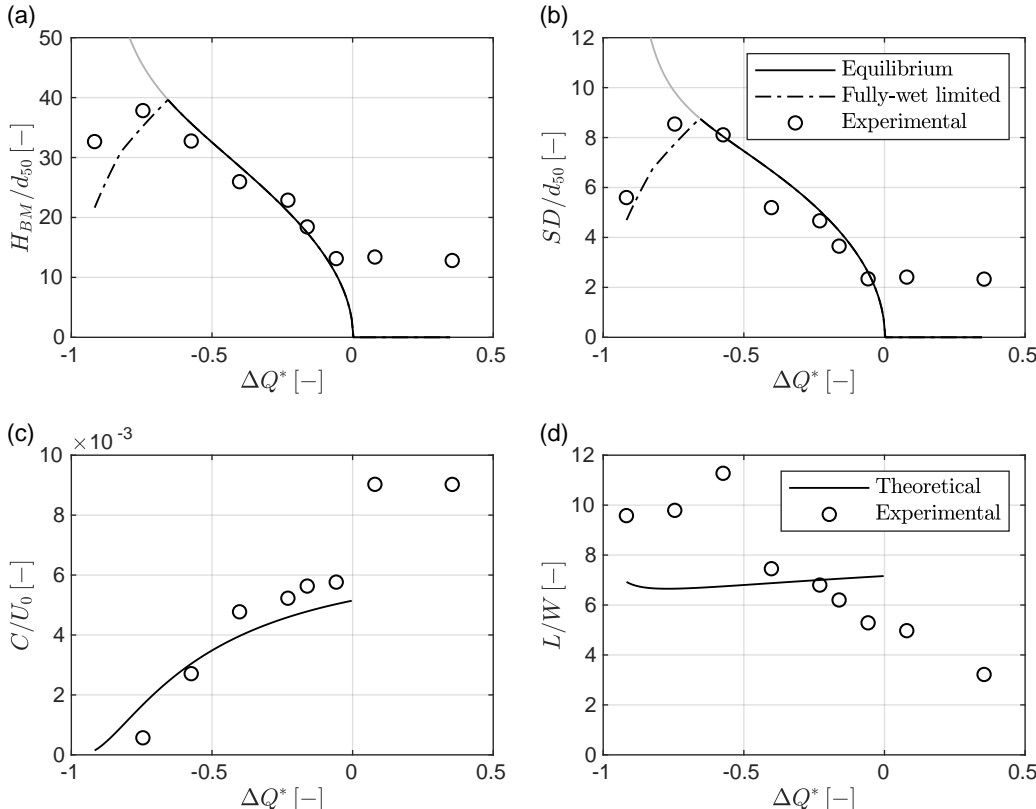

**Figure 5.** Dimensionless bar parameters as a function of the scaled discharge, from theory (lines) and experiments (markers). (a) and (b): height and standard deviation of bed elevation distribution (scaled with the water depth $D_0$), with solid line indicating equilibrium conditions and dash-dot line representing the bar height limited by the fully-wet condition. (c) and (d): bar migration rate and wavelength (scaled with the flow velocity $U_0$ and the channel width $W$, respectively).

period in one channel width. Harmonic components with $n = 0$ and $m = 0$ are constant along the $x$ and $y$ axis, respectively. The

component $A_{00}$, which represents a horizontal plane, has amplitude equal to zero, because the original DEM was normalized by removing the mean.

As revealed by their total energy content, $E_n$, the most important components of the spectrum are those with longitudinal mode $n = 1$. Specifically, alternate bars are dominated by the component $A_{11}$, which represents a double sinusoidal bed deformation. Components with $n = 1$ and higher, odd transverse mode $m$, such as the $A_{13}$ and the $A_{15}$, also appear in the spectrum,

contributing to the deviation of the cross section from a purely sinusoidal variation to a more complex (but still antisymmetric) shape.

However, components with longitudinal modes $n = 0$ and $n = 2$ are also relevant. The longitudinal mode $n = 0$ is dominated by the component $A_{02}$, which represents a sinusoidal symmetric bed deformation that is constant in $x$, while the components

Earth **Surface**
**Dynamics**
Discussions

**Figure 6.** Maps of ensemble bars, representing variation of the average bar topography for increasing values of discharge. Spatial coordinates $(x,y)$ are normalized with respect to the bar wavelength ($L$) and the channel width ($W$). Contour spacing is $5\,\mathrm{mm}$ (upper panels) and $2\,\mathrm{mm}$ (lower panels), with the thicker contour indicating the mean (i.e. zero) elevation and white contours representing negative elevation values. Flow is from top to bottom.



Earth **Surface**
**Dynamics**
Discussions

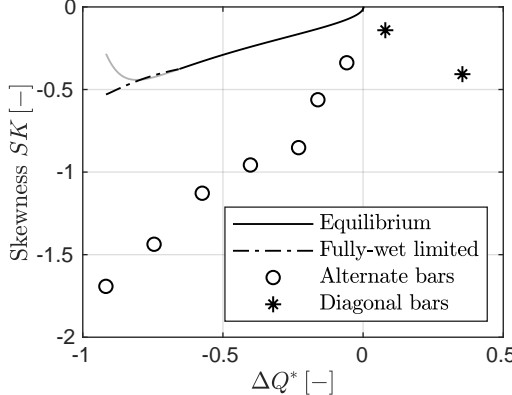

**Figure 7.** Skewness of the bed elevation distribution as a function of the scaled discharge. Markers indicate the skewness of the experimental ensemble bars; lines illustrate results from the weakly nonlinear theory, with the dash-dot line referring to the solution obtained by limiting the bar growth to the fully-wet condition.

with longitudinal mode $n = 2$ include a number of (even) transverse modes (i.e. $A_{22}$, $A_{24}$, which represent a symmetric bed
deformation that completes two periods in one bar wavelength (see Fig. 9 for a schematic representation).

All $m = 0$ harmonics, which represent a purely longitudinal bed deformation, turn out to be negligible, showing that the transversally averaged bed elevation is nearly zero for all the cross sections. Analogously, components with $n = 0$ and odd transverse mode $m$ are also vanishingly small, which implies that on average the bed structure does not exhibit any asymmetry with respect to the channel axis.

For all tested conditions, the Fourier spectrum exhibits a clear checkerboard pattern, where at least $98\%$ of the energy is contained in even-even and odd-odd modes, while other harmonics have negligible power (on average $0.7\%$). This distinctive pattern indicates that despite their morphological complexity, both alternate and diagonal bars are "purely alternate", in the sense that the second half of the bar is nearly identical to the first half but mirrored across the channel axis.

We note that the above results are valid in general, regardless of the value of flow discharge. However, relevant variations
of the Fourier spectrum composition occur when changing $Q$. The amplitude of the four dominant components $A_{11}$, $A_{13}$, $A_{20}$ and $A_{22}$ is illustrated in Fig. 9 as a function of the dimensionless discharge previously defined in Eq. (9). The amplitude of the fundamental harmonic, $A_{11}$, which is illustrated in Fig. 9a, closely follows the trend observed for bar height parameters (see 3a and 3b), with maximum values for $Q = 1.0$ and $1.5\,\mathrm{l\,s^{-1}}$, a steady decrease up to $Q = 3.0\,\mathrm{l\,s^{-1}}$ and lower, almost constant values afterwards. This is not surprising, as the $A_{11}$ is the dominant component of the bed topography, which therefore mostly
determines the bar height.

To quantify the shape of the bars, regardless of their absolute height, we then refer to the relative amplitude of the Fourier modes, given as a proportion of the amplitude $|A_{11}|$ as illustrated in Fig. 9b,c,d. Excluding the first case ($Q = 0.5\,\mathrm{l\,s^{-1}}$), the trend observed for alternate bars is rather clear, with a decreasing importance of both the symmetric component $A_{02}$ and the asymmetric component $A_{13}$, from amplitudes of about 60–70$\%$ of $|A_{11}|$ to significantly smaller values when approaching the



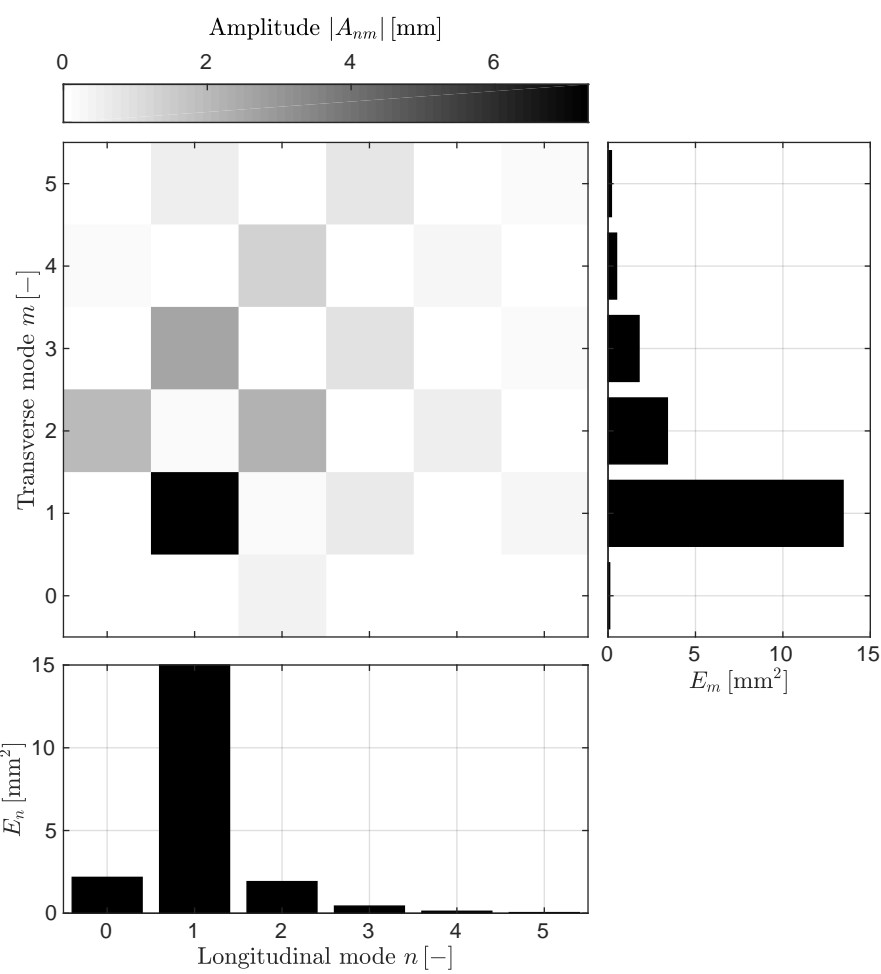

**Figure 8.** Amplitude of the first $6 \times 6$ longitudinal and transverse modes for the ensemble bar at $Q = 2.5 \, \mathrm{l \, s^{-1}}$. Histograms on the bottom and on the left also reports the total energy $E_n$ and $E_m$, representing the variance of the bed elevation associated with all the components having longitudinal mode $n$ and transverse mode $m$, respectively.

Earth **Surface**
Dynamics
Discussions

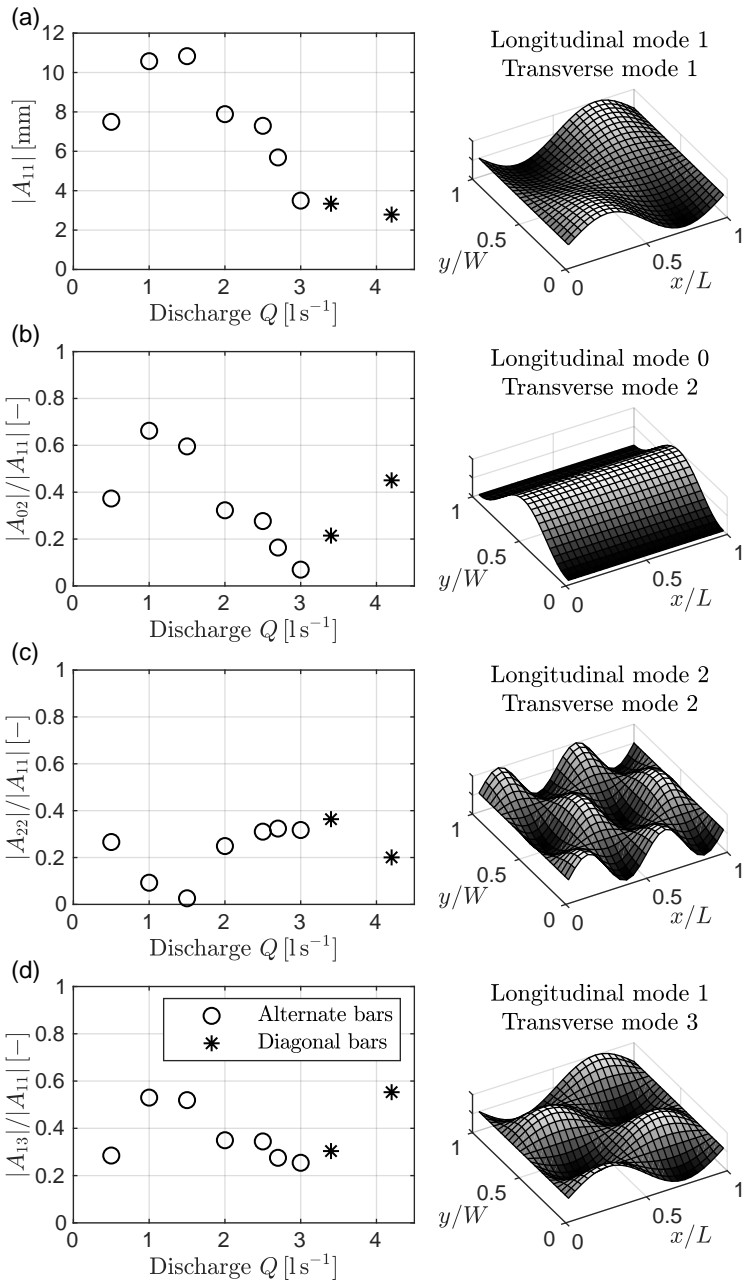

**Figure 9.** Amplitude of the main Fourier components depending on discharge. (a): amplitude of the fundamental harmonic ($A_{11}$); (b), (c) and (d): ratio between the amplitudes $A_{02}$, $A_{22}$, $A_{13}$ and the fundamental. The 3-D plots on the right illustrate the bed deformation associated with each Fourier component.





critical threshold $Q_{cr}$. Specifically, the amplitude of the component $A_{02}$ decreases by about an order of magnitude. However, the component $A_{22}$ shows an inverse (i.e. increasing) trend, from the nearly vanishing amplitude observed at $Q = 1.0$ and $1.51 \mathrm{s}^{-1}$ to values of about $1/3$ of $|A_{11}|$ when approaching $Q_{cr}$.

The weakly nonlinear theory of Colombini et al. (1987) resolves the firsts $2 \times 2$ modes of the Fourier spectrum, thus allowing for calculating the expected variation of the main components illustrated in Fig. 9, except for the $A_{13}$. The amplitude of the

fundamental harmonic $A_{11}$ (Fig. 10b) is fairly well reproduced, at least for values of the flow discharge that do not differ much from the critical threshold $Q_{cr}$, that is where the theory is expected to work best. However, for lower discharge values the theoretical curve clearly overestimates the measured data, and the predicted equilibrium amplitude diverges as discharge approaches the threshold value $Q_i$. In this case, as also noticed earlier, an approximate solution can be derived by assuming that the bar growth is limited by the fully-wet condition.

To investigate the overall importance of the $m = 2$ components with respect to the fundamental, we first analyse the sum of the absolute values of the coefficients $A_{02}$ and $A_{22}$, scaled with the amplitude of the fundamental harmonic $A_{11}$. As illustrated in Fig. 10b, this metric tends to decrease with discharge, theoretically approaching zero near the critical conditions (i.e at $\Delta Q^* \simeq 0$). Despite its limited capability to quantify the fully nonlinear interactions, the theory allows for a proper estimation of the observed values. However, the main difference between theory and experimental data lies in the relative amplitude of

the individual $m = 2$ components, $A_{02}$ and $A_{22}$. As illustrated in Fig. 10c, the values of the ratio $A_{02}/A_{11}$ are strongly underestimated by the theory, with experimental values being roughly four times their theoretical counterpart. On the other hand, values of the ratio $A_{22}/A_{11}$ reported in Fig. 10d are significantly overestimated. This indicates that the mode-2 component is not dominated by the presence of regular, periodic central bars (see map of Fig. 9c), as suggested by the theory, but it is mainly associated with a bell-shaped distortion of the average cross section (Fujita and Muramoto, 1985), as represented in Fig. 9b.

Finally, it is worth noting that the ratio $A_{22}/A_{11}$, illustrated in Fig. 10d, does not tend to zero as predicted by the theory. From a morphological point of view this implies that for $Q \to Q_{cr}$, while "theoretical" bars tend to become purely sinusoidal ($A_{11}$ component only) as the solution approaches its linear limit, observed bars retain a certain degree of non-linearity and keep a $m = 2$ component that derives from the presence of clear diagonal fronts (see Fig. 6).



Earth **Surface**
**Dynamics**
Discussions

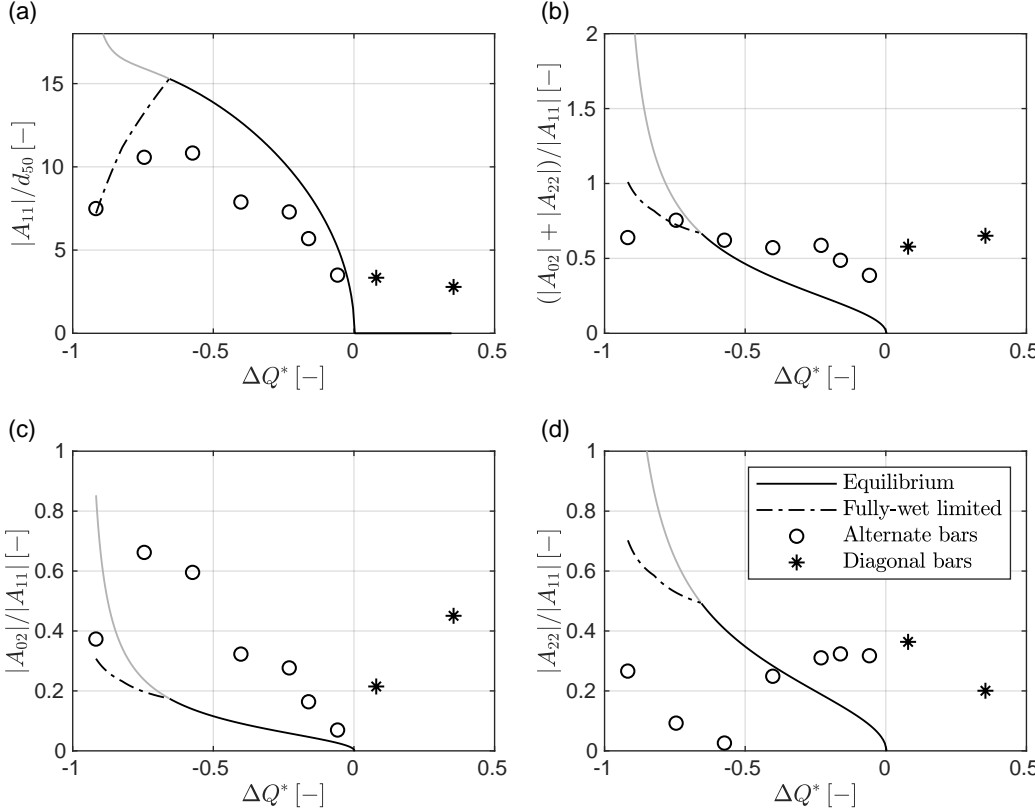

**Figure 10.** Amplitude of the main the Fourier components depending on the scaled discharge, from theory (lines) and experiments (markers). Panel (a): amplitude of the fundamental component $A_{11}$, scaled with the grain size $d_{50}$. Panels (b,c,d): amplitude of the $m = 2$ components, scaled with $|A_{11}|$, with (b) reporting the sum of the absolute values of the coefficients $A_{02}$ and $A_{22}$; (c) and (d) referring to the individual components $A_{02}$ and $A_{22}$. Solid line indicates theoretical results at equilibrium, while dash-dot line indicates theoretical results obtained by limiting the bar growth to the fully-wet condition.

## 4 Discussion

### 4.1 Discharge and bar height

Experimental data reveal that bar height and relief generally decrease with discharge, and are therefore inversely correlated with the sediment transport rate. This finding, which at a first sight may appear counterintuitive, is a direct consequence of the decrease of channel aspect ratio for progressively higher flows that is typical of single-thread rivers. This implies that the largest bedforms tend to develop under moderate flow conditions, where discharge is high enough to mobilize the bed material and at the same time is sufficiently low with respect to the critical discharge for bar formation, $Q_{cr}$.





The decreasing bed relief with discharge is expected to have a direct impact on the sediment transport rate. Specifically, for relatively low values of discharge the presence of bars can promote a transversal variability of the Shields number, which leads to a net increase of the sediment transport rate with respect to equivalent, flat bed conditions (e.g., Paola, 1996; Francalanci et al., 2012). This effect can mitigate the reduction of the average transport rate when decreasing discharge, thus making the

sediment rating curve more linear (Ferguson, 2003; Redolfi et al., 2016).

Our results reveal that the weakly nonlinear model allows for reproducing both qualitatively and quantitatively the observed bar height. Despite the calibration of the parameter $r$, this is a significant result, as it highlights the capacity of the theoretical model to accurately capture the sharp decreasing trend of bar height from intermediate values of discharge to the critical threshold.

However, the variation of the bar height with discharge is not everywhere monotone, as when discharge becomes relatively low, bars tend to emerge, and their relief tends to reduce. In these conditions the equilibrium amplitude predicted by the Colombini et al. (1987) model is clearly unphysical. This is not surprising, as the theory assumes a simply connected domain, where the bed is fully submerged. To mitigate this issue, we propose a modified curve for the bar height, obtained by limiting the growth of the bars by the fully-wet condition.

Situations where bars emerge are expected to be more important for wider channels, due to the larger range of discharge states between the threshold of incipient sediment transport and the fully-wet threshold. Specifically, as the channel width-to-depth ratio grows, the equilibrium becomes increasingly complex, ultimately leading to wandering and braiding channels (see Ashmore, 2013; Garcia Lugo et al., 2015; Redolfi et al., 2016).

### 4.2   Discharge and bar shape

The definition of suitable metrics for quantifying variations of the bar shape allows us to highlight how the shape of alternate bars at equilibrium changes with discharge.

The weakly nonlinear theory of Colombini et al. (1987), as well as the bar predictor of Crosato and Mosselman (2009), suggest that when discharge is relatively small (i.e. high width-to-depth ratio) the channel tends to form regular, periodic central bars (also called double-row bars, see Ikeda, 1984; Crosato and Mosselman, 2020) where scour and deposition are

equally distributed between the center of the channel and the area near the banks.

However, as also evident from existing laboratory and numerical data (e.g., Fujita and Muramoto, 1985; Garcia Lugo et al., 2015; Qian et al., 2017; Cordier et al., 2019) it is clear that deposition preferentially occurs near the center of the channel (mid-channel bars), while deep pools are mostly concentrated near the banks. This produces a bell-shaped distortion of the average cross section, which gives a Fourier component $A_{02}$ that at low flows is by far more important than the $A_{22}$ component.

As highlighted by Colombini and Tubino (1991) this behavior can be explained by fully taking into account the nonlinear effects, which tend to be progressively more important when the channel aspect ratio increases. Moreover, mid-channel bars are typically not symmetric with respect to the channel centerline, but they often appear as compound bars, with the water flow mainly concentrating near the banks and sometimes cutting the entire channel width through the formation of channel bifurcations (e.g., Schuurman and Kleinhans, 2015; Duró et al., 2016).





In experimental modelling of wandering and braided rivers (e.g., Ashmore, 1982; Garcia Lugo et al., 2015) the tendency of the channels to "stick to the banks" is often considered as a side effect of the physical model. However, this may be not necessarily the result of scaling issue, nor a consequence of the low roughness of the banks, but it could be associated with a natural tendency of the flow to follow the banks when they are sufficiently straight, with the bars mainly occupying the mid part of the channel.

The Fourier analysis also reveals that the component $A_{22}$ (as well as the $A_{13}$) does not vanish when approaching the critical conditions, as the theory predicts. Considering that near $Q_{cr}$ the weakly nonlinear analysis should provide an accurate solution of the shallow water and Exner equations, the mismatch is likely to originate from the model equations themselves. Specifically, this may be related to three-dimensional effects, which could be locally important in determining the formation of relatively steep bar fronts that mark a significant difference with respect to the theoretical, sinusoidal bed structure.

### 375  4.3   The observed transition between different types of bars

Experimental observations presented in this paper provide detailed information on the relationship between bar characteristics and discharge, while other relevant channel properties, such as grain size and slope, are kept constant. Within the tested flow range, bars exhibit a variety of sizes and shapes and pass smoothly from one shape to the other as discharge increases. On the basis of their geometrical properties and migration rate it is possible to identify four main types of bars:

1. At low flows, when channel aspect ratio is high, alternate bars are very irregular, and the channel tends to switch to a more complex, wandering morphology. Sediment transport occurs on a limited portion of the bed, and the bed evolution is not dominated by the downstream migration of bar fronts, but rather by lateral erosion and cutoffs. This kind of bars can be associated to conditions where the top of the bars emerges, so that the bed is not fully wet ($W_w < W$, see Table 1). The emersion limits the possibility of bars to grow in height, and consequently restricts the bed relief.

2. At low to intermediate flows, bars are dominant and relief is high. Their transverse shape is highly asymmetric, with narrow, deep, elongated pools and high, flat bar tops occupying a large proportion of the cross section, so that the elevation along the centerline of the channel is always above the median detrended elevation. The distribution of elevation is strongly negatively skewed, and the Fourier components $A_{02}$ (symmetrical deformation) and $A_{13}$ (asymmetrical deformation) are relatively strong. Bar fronts are clearly delineated, steep and almost orthogonal to channel banks. Immediately downstream of fronts, where the deepest pools are located, there is no sediment in motion. Moreover, bar migration is slow and the wavelength is significantly higher than the theoretically predicted values.

    3. At intermediate to high flows, relief and bar wavelength decrease with increasing discharge, bar fronts become curved and oblique. The bed elevation distribution is less skewed and higher-order components of the Fourier spectrum become less relevant with respect to the fundamental harmonic $A_{11}$. As deep pools tend to disappear, sediment motion occurs on 395    the entire channel surface. This kind of bedforms represents the typical shape of alternate bars (i.e. that sketched in Fig. 1a), and shows a very close match with theoretical predictions in terms of bar height, wavelength and migration rate.





4. Finally, at high flows we observe the formation of diagonal bars. Despite preserving an alternate shape, diagonal bars are rather different from alternate bars in terms of both geometrical properties and formation mechanism. The height of these bedforms is small and largely independent from discharge, and their elevation distribution is almost symmetri-cal. Diagonal bars are relatively short (less than five channel widths), with oblique, almost straight fronts that migrate downstream at high speed. They are observed outside the range of alternate bar formation (i.e. for $Q > Q_{cr}$) and should be regarded as a distinct kind of bedforms that can be interpreted as three-dimensional oblique dunes (Colombini and Stocchino, 2012). Discharge values for which diagonal bars develop are consistent with the empirical criterion proposed by Jaeggi (1984).

The three-dimensional character of the flow field is fundamental for explaining the morphology of diagonal bars. Specifically, when $Q > Q_{cr}$ the two-dimensional, depth-averaged model of Colombini et al. (1987) would predict plane-bed conditions (no bars), while a three-dimensional, non-hydrostatic analysis is needed to reproduce the observed formation of diagonal bars (Colombini and Stocchino, 2012). For this reason, all depth-averaged numerical models for alternate bars (e.g., Crosato et al., 2011; Siviglia et al., 2013; Qian et al., 2017; Cordier et al., 2019) are likely suffering the same limitation. Since diagonal bars are of small amplitude, they can be expected to have a negligible effect on sediment transport, flow resistance and interaction with other bedforms; however, attention should be paid in the interpretation of numerical results and in their comparison with field and laboratory observations.

From a visual inspection of the topographies illustrated in Fig. 2, it is evident that bars forming at $Q > 2.0\,\mathrm{l\,s}^{-1}$ are not spatially uniform, but they grow in the initial part of the channel, before adapting to fully developed conditions. This behavior has been often observed by laboratory and numerical experiments (Fujita and Muramoto, 1985; Defina, 2003; Nicholas, 2010; Qian et al., 2017) and has been associated to the fact that bar formation needs to be triggered by small perturbations, whose effect tends to propagate the downstream direction in the form of wave packages (i.e. trains of bars). Specifically, the spatial adaptation is probably a consequence of the convective (rather than absolute) nature of bar instability highlighted by Federici and Seminara (2003), which implies that the effect of local perturbations tends to be convected downstream rather than being spread throughout the whole domain.

On the basis of theoretical results, it is possible to define an additional threshold value of discharge, corresponding to condi-tions where the channel aspect ratio $\beta$ equals the resonant value $\beta_R$, originally defined by Blondeaux and Seminara (1985) (see Table 1). We name this threshold value "resonant discharge", which assumes a value $Q_{res} = 1.94\,\mathrm{l\,s}^{-1}$. Although not directly affecting the theoretical solution for free migrating bars, the resonant threshold is fundamental for defining the propagation of morphological effects that can be generated by any flow disturbance (e.g., that associated with boundary conditions). Specifi-cally, as first highlighted by Zolezzi and Seminara (2001), under sub-resonant conditions (i.e $Q > Q_{res}$) morphological effects tend to manifest themselves downstream of the disturbance, while an upstream propagation is possible in the super-resonant regime (i.e. when $Q < Q_{res}$).

The different behavior of bars observed at relatively low flows, which tend to be well-developed along the entire flume (see Fig. 2), may be associated with the super-resonant character of the experiments. In this case the possible upstream propagation





of the morphological information may favour an upstream diffusion of the bed instability, which can therefore reach the initial part of the channel.

## 4.4 The alternate nature of both alternate bars and diagonal bars

The checkerboard pattern of the Fourier spectra indicates that both alternate and diagonal bars are "purely alternate", in the sense that the elevation map of the upstream half wavelength is nearly identical to the downstream half but mirrored along the channel centerline. Note that this does not imply a point symmetry with respect to the center of the bar ($y = W/2$ and $x = L/2$), but rather a switching of the same erosion and deposition pattern between the two sides of the channel. Interestingly, this is valid even for the ensemble bars at the lowest discharge ($Q = 0.5 \, \mathrm{l \, s^{-1}}$), despite the complexity of the bed topography that can be appreciated in Fig. 2.

This particular pattern is intrinsically linked to the bar formation mechanisms. To some extent, both alternate and diagonal bars can be considered as "free bars", in the sense that they both arise from an autogenic, three-dimensional instability of the erodible bed. This kind of instability does not break the overall symmetry of the problem; therefore, if a deposition patch tends to form near one bank, a similar feature should appear somewhere else, but on the opposite side of the channel. This suggests that if periodic, three-dimensional bedforms develop, they should follow an alternate pattern, at least in an average, statistical sense.

From a mathematical point of view, the checkerboard pattern can be explained by considering that free bars tends to initially appear as a bed deformation having a double sinusoidal shape ($A_{11}$ component only), while as they grow nonlinear interactions gives rise to the second-order, $A_{00}$, $A_{02}$ $A_{20}$, $A_{22}$ harmonics (see Colombini et al., 1987; Bertagni and Camporeale, 2018). Extending the analysis to higher order of approximation would give other even-even modes, but no odd-odd modes.

Finally, it is worth noting that the dominance of the even-even and the odd modes also has an experimental significance, as it indicates that: (i) there are not systematic trends that may be associated with channel asymmetries (i.e. in the initial bed levelling); (ii) random effects resulting from measurement errors, experimental imperfections or associated with the intrinsic stochasticity of sediment transport processes are not significantly affecting the shape of the ensemble bar.

## 4.5 Suitable metrics for quantifying bar height and relief

The laboratory dataset used for this work allowed for the comparison of a number of methods and metrics to characterize bar height and relief.

Historically, interest in the quantification of bedform height arose from their influence on human activities and interaction with artificial structures (Jaeggi, 1984). Therefore, maximum scour and deposition were the most relevant parameters to evaluate the risk of levee instability and levee overtopping, respectively. However, metrics of bar height based on maxima and minima ($H_{BM}$ and $H_B$) are highly sensitive to measurement errors and uncertainties that may derive from the presence of vegetation on the bar top and from the difficulty to measure the bottom elevation in deep pools. Moreover, the estimation of both $H_{BM}$ and $H_B$ requires the identification of individual bar units, which introduces potential sources of uncertainties and limits its application to bed configurations where a dominant longitudinal wavelength is clearly recognizable.





On the other hand, $SD$ and $BRI$ are robust indices that do not depend on extreme values of elevation but on the entire bed
elevation distribution. Moreover, these bed relief metrics can be applied to a range of different morphologies, thus allowing
for comparisons between bars and other bedforms. Since $SD$ and $BRI$ show the same trend observed for $H_{BM}$ and $H_B$, the
formers can provide better data when the purpose is not to quantify the maximum scour and deposition position but rather to
measure morphological trajectories and to compare study cases with experimental and numerical simulations.

It is also important to note that metrics based on the comparison of elevation values at different longitudinal positions (i.e.
$H_{BM}$ and $SD$) require detrending the bed elevation by removing an average slope that is often not obvious to define. Our
experiments show that results are very similar when considering instead the cross section based indices $H_B$ and $BRI$, with the
advantage that they are fully independent on how the average slope is detrended. This is linked to the presence of deep, small
pools and large, flat bar tops. Cross-sectional relief is more strongly influenced by the former and the maximum elevation along
the cross section where the lowest point is located is not very different from the highest point of the entire bar.

## 475  5   Conclusions

We explored how the equilibrium properties of free, migrating alternate bars depend on water discharge through a series of
laboratory experiments, where width, channel slope and bed material were kept constant. A proper definition of the most
suitable metrics, the analysis of the experimental results, and the comparison with existing theoretical models, allow us to draw
the following conclusions:

1. the equilibrium bar height generally decreases with discharge. However at low flows, when bars start emerging from the
water surface, an opposite trend is observed, which implies that moderate flows are mainly responsible of the formation
of large alternate bedforms;

2. the shape of alternate bars significantly changes with discharge, with relatively low flow conditions characterized by a
high negative skewness of the bed elevation distribution and an important contribution of the higher-order Fourier modes
with respect to the fundamental harmonic;

3. at low discharge, when the width-to-depth ratio is relatively high, the mode-2 Fourier components tend to become
increasingly important. However, the channel does not tend to develop a regular central bars but rather a bell-shaped
distortion of the average cross section, with deposition preferentially occurring near the center of the channel (mid-
channel bars), and scour pools mainly located near the banks;

4. the significant variations of the bar morphology and the associated metrics, allows for identifying four main types of
bars, which can be associate to different flow conditions with respect to the relevant morphodynamic thresholds;

5. the weakly nonlinear theory allows for a satisfactory prediction of bar height and migration speed, while their capability
to reproduce bar shape is limited to a qualitative analysis. Moreover, limiting the bar growth to the fully-wet condition





allows for correcting the theoretical predictions at low values of discharge, for which alternate bars tend to emerge from
the water surface;

6. the transition from alternate bar morphology to plane-bed configuration that is expected when discharge exceeds the
critical threshold $Q_{cr}$ is not sharp, due to the formation of diagonal bars, which can be regarded as three-dimensional
oblique dunes;

7. the definition of ensemble bars that represent the average bar topography enables us to clearly highlight the "purely
alternate" character of both alternate bars and diagonal bars, which manifests itself as checkerboard pattern of the Fourier
spectrum.

Overall, this work provide fundamental information for designing laboratory experiments and numerical simulations, for
predicting the bar evolution in different scenario of possible hydrological alterations, and for interpreting the observed mor-
phological changes depending on channel conditions and flow history.

*Code and data availability.* A Matlab code for the for the computation of the critical and resonant conditions (Redolfi et al., 2019) is avail-
able at https://bitbucket.org/Marco_Redolfi/bars_res-crit, while a Matlab function for the Fourier analysis of bed topographies is provided at
https://bitbucket.org/Marco_Redolfi/fourier_transform_bars. Laboratory data are available at https://doi.org/10.5281/zenodo.3746619 (Wel-
ber et al., 2020).

## Appendix A: Calculation of Fourier coefficients

Here we detail the procedure needed to expand the signal in the form of Equation (5), and to calculate the associated coefficients
$A_{nm}$. We start by considering a generic real-valued, two-dimensional signal $f_{jk}$, defined on a regular grid of $J \times K$ points,
whose indexes $j$ and $k$ run from 0 to $J-1$ and from 0 to $K-1$, respectively. The two-dimensional, discrete Fourier transform
allows for expressing the signal as follows:

$$f_{jk} = \sum_{m=0}^{J-1} \sum_{n=0}^{K-1} F_{nm} \exp\left[2\pi i n \frac{j}{J} + 2\pi i m \frac{k}{K}\right],$$
(A1)

where $i$ is the imaginary unit and the complex Fourier coefficients $F_{nm}$ can be calculated through standard Fast Fourier
Transform algorithms (e.g., the Matlab FFT2 function).

Considering the $L \times W$ domain illustrated in Figure A1, with the system of reference $(x, y)$ originating at the lower left
corner, the coordinates of the grid points can be determined as $x = (j + 0.5)\,dx$ and $y = (k + 0.5)\,dy$, where $dx = L/J$ and
$dy = W/K$ are the grid spacing in the longitudinal and transverse directions, respectively. In this system of reference, the
Fourier expansion of the signal can be expressed as:

$$f_{jk} = \sum_{m=0}^{J-1} \sum_{n=0}^{K-1} F_{nm}^* \exp\left[2\pi i n \frac{x}{L} + 2\pi i m \frac{y}{W}\right],$$
(A2)





whose Fourier coefficients can be readily computed as:

$$F_{nm}^* = F_{nm} \exp\left[-\pi i \left(\frac{n}{J} + \frac{m}{K}\right)\right]. \tag{A3}$$

Equation (A2) contains both sines and cosines in both the $x$ and $y$ directions, as evident when expanding the complex

exponential by means of the Euler's identity:

$$f_{jk} = \sum_{m=0}^{J-1}\sum_{n=0}^{K-1} F_{nm}^* \left[\cos\left(2\pi n\frac{x}{L}\right)\cos\left(2\pi m\frac{y}{W}\right) - \sin\left(2\pi n\frac{x}{L}\right)\sin\left(2\pi m\frac{y}{W}\right) + \right.$$
$$\left. i\cos\left(2\pi n\frac{x}{L}\right)\sin\left(2\pi m\frac{y}{W}\right) + i\sin\left(2\pi n\frac{x}{L}\right)\cos\left(2\pi m\frac{y}{W}\right)\right]. \tag{A4}$$

Here we are interested to obtain an expression where, consistently with theoretical analyses (e.g., Colombini et al., 1987),

only cosines appear in the transverse structure. To obtain this expression, we can start by considering that the signal $f_{ij}$ can

be represented in a different way by adding virtual, external points at which arbitrary values are assigned to the signal. This

technique is rather common in signal analysis to obtain a different Fourier representation of the same signal: for example, a

zero padding is often used to increase the wavelengths of the fundamental harmonic. Specifically, if we extend the grid by

adding $K$ virtual points in the $y$-direction as illustrated in Figure A1, and we compute the Fourier transform as detailed above,

we obtain the following expression:

$$f_{jk} = \sum_{m=0}^{J-1}\sum_{n=0}^{2K-1} F_{nm}^* \exp\left[2\pi i n\frac{x}{L}x + 2\pi i m\frac{y}{2W}\right], \tag{A5}$$

which is similar to Eq. (A2), except for the transverse wavelength of the fundamental harmonic being twice the channel width

($2W$). The key to eliminate the sine components along the $y$-direction, is to properly assign the values of $f_{ij}$ at the virtual

points. Specifically, if the signal is mirrored with respect to the $y = W$ axis, the sum all the terms containing $\sin(y)$ identically

vanishes, so that the Fourier expansion (A5) can be written as follows:

$$f_{jk} = \sum_{m=0}^{J-1}\sum_{n=0}^{2K-1} F_{nm}^* \cos\left(2\pi m\frac{y}{W}\right)\left[\cos\left(2\pi n\frac{x}{L}\right) + i\sin\left(2\pi n\frac{x}{L}\right)\right]. \tag{A6}$$

Equation (A6) contains redundant information, as components actually having an identical structure appear more than once

in the sum. Specifically, It is possible to demonstrate that only $M \times N$ components are needed to exactly represent a real signal,

where $N = K$ and $M$ equals $J/2 + 1$ or $(J+1)/2$, depending on $J$ being an even or an odd number, respectively. Therefore,

a proper definition of the coefficients $A_{nm}$ allows for expanding the signal $f_{ij}$ in the following parsimonious way:

$$f_{jk} = \sum_{m=0}^{M-1}\sum_{n=0}^{N-1} \cos\left(2\pi m\frac{y}{2W}\right) Re\left\{A_{nm}\left[\cos\left(2\pi n\frac{x}{L}\right) + i\sin\left(2\pi n\frac{x}{L}\right)\right]\right\}, \tag{A7}$$

which can be equivalently written in the form of Equation (5) after expressing the complex coefficients in terms on their

amplitude and phase:

$$A_{nm} = |A_{nm}| \exp(i\,\phi_{nm}). \tag{A8}$$





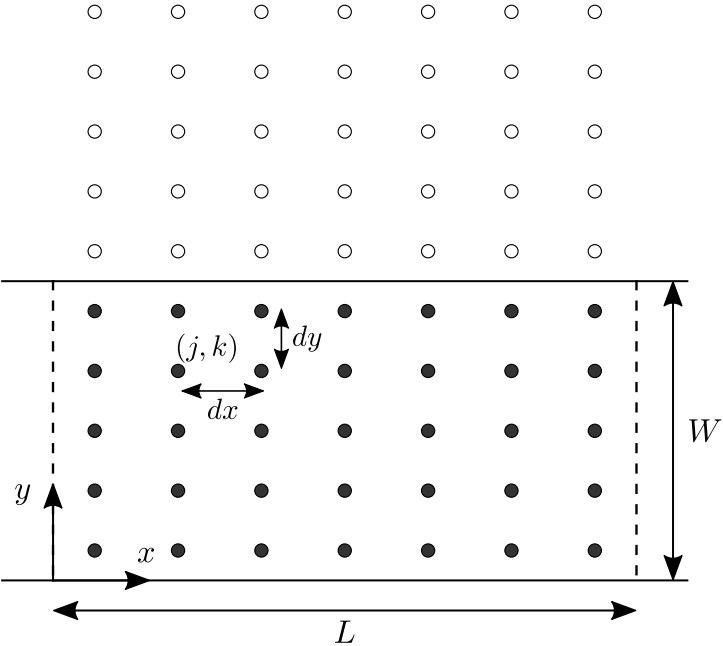

**Figure A1.** Illustration of the grid used to discretize a domain of size $L \times W$, where $W$ is the channel width. Grid points are identified by the indexes $j$ and $k$ ($x$ and $y$ direction, respectively), and are equally spaced at intervals $dx$ and $dy$. The original grid contains $J \times K$ points (closed circles), while the extended grid, obtained by adding virtual external points (open circles), is formed by $J \times 2K$ points that cover a total width $2W$.

The Fourier coefficients $A_{mn}$ can be directly derived from the $F_{nm}^*$ coefficients, and can be computed for a generic $f_{jk}$ signal using the Matlab code we made available at https://bitbucket.org/Marco_Redolfi/fourier_transform_bars.

*Author contributions.* MR analyzed the data and wrote the manuscript; MW designed and performed the experiments, processed data and wrote part of the manuscript; MC performed the calculations by means on the weakly nonlinear theory; MT supervised the work; WB designed the experiments and revised the paper. All authors contributed in the interpretation of the results.

*Competing interests.* The authors declare that they have no conflict of interest.

*Acknowledgements.* The research was funded by the project: GLORI - Glaciers-to-rivers sediment transfer in Alpine basins (Autonomous Province of Bozen-Bolzano).



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
