# Peer review of "Morphometric properties of alternate bars and water discharge: a laboratory investigation"

_Earth Surface Dynamics, 2020_

## Referee Comment (RC1) · Christopher Paola (Referee) · 26 May 2020

The paper is a straightforward, nicely presented, experimental and theoretical study of the effect on channel bars of systematically increasing water discharge. Because the experiments use a conventional vertical-wall flume, the increase in discharge translates to increasing depth and shear stress at constant width and slope. The result is a transition from emergent bars that lead to a rough meandering pattern through conventional alternate bars to a form of diagonal bar that seems to be a transition to dunes. The bars generally get lower and shorter, and move faster, as the discharge increases. The paper also shows that many of the changes are reasonably well predicted by the weakly nonlinear bar theory proposed in 1987 by Colombini and colleagues.

[Figure]

Apart from the direct scientific findings, two meta-contributions of the paper that I particularly like are, first, a nice demonstration of a mechanistic theory applying to conditions well outside those for which it was formulated; and second, an especially clear and teachable example of the application of Fourier modes in morphodynamic theory (Fig. 5). All in all, this is a paper that asks relatively little of its reviewers. So perhaps the most useful thing I can do, apart from minor technical suggestions, is to pose some questions that might provide a starting point for additional discussion of the paper's findings:

1. Are there any visible effects of the near-critical Froude numbers that characterize all of the runs (Fr values roughly 0.9 – 1.2, Table 1)?

2. Do any of the observed bars show flow separation, and if so, what effects do the authors think the separation, which is clearly not part of a weakly nonlinear theory, has on bar dynamics and sediment transport?

3. From the parameter values in Table 1, it appears that the experiments were all run for bedload-dominated conditions, and with a relatively high relative roughness based on the grain size. Overall, the experimental conditions are representative of gravel-bed rather than sand-bed rivers. In that context it's interesting to consider the transition to dunes hinted at in the higher-discharge runs, since most natural, alluvial gravel rivers do not have dunes, while dunes are the predominant bedform in sand-bed rivers, often superimposed on bars. The intermediate case, diagonal bars, in the high-discharge runs appears to be relatively unusual in natural channels. Yet they do not appear to require unusual conditions to form in the experiments. Do the authors have any thoughts as to what aspects of self-organization in natural channels might discourage diagonal bars or other transitional-dune forms from developing? Or have they been overlooked or misclassified? 4. On the same theme of diagonal bars, this set of experiments seems to beg for a follow-up in which the aspect ratio is varied over an even wider range, allowing the slope to vary also, so that one could observe the complete transition from emergent bars and an inset meandering channel (the lowest

discharge in this series) to a state of fully developed dunes with a height scaled to, but only a fraction of, the flow depth (the logical extension of the high-discharge end of this series). Do the authors think this would be useful, and/or have any other comments on the relation between bar and bedform dynamics?

Minor technical comments: I would like to see the following quantities added to Table 1 for each discharge: spatial mean shear stress; Shields parameter based on that stress; Rouse number. I believe that Esurf uses UK standard spelling, in which case 'center' should be changed to 'centre' throughout the paper. 29/30 'this kind of bedforms' should be 'this kind of bedform'. I would also suggest that, although not technically incorrect, many people use 'bedform' only for features like ripples and dunes whose vertical scale is small compared to the depth. A more neutral term like 'bed morphology' would be less likely to cause confusion. 49 'Fredsoe, 1978' It seems to me that Parker (1976) made this point clearly, though earlier, and should be added to the citation here. 57 'manly' to 'mainly' 122 'cross sections' This again is not technically incorrect but people may interpret 'cross' to imply 'transverse' so this might clarified to 'longitudinal sections' 151 'amplitude... provides' to 'amplitudes...provide' 157 'requires to specify' to 'requires that we specify'. Additionally, I assume that the authors mean that in the original Colombini et al. theory these closure relations were not specified, so that equations 6 − 8 represent choices made by the authors of the current paper. If this is the case, it would be better not to use the past tense in describing these choices (e.g. in line 168 'was modeled'), since they are part of this paper. So for example, in line 168, change to something like 'We model the effect...' But it seems to me that Colombini et al. included a similar closure for the lateral slope effect, so it's not clear which aspects of equation 8 are different (only the value of r?) between this paper and the original theory. 184 'becomes' to 'become' 202 'Noteworthy,' to 'It is noteworthy that' 234 'to' to 'for' 250/1 'ensemble bar shape' Computation of these for each discharge is one of the more interesting data-analysis techniques used in the paper. It would be nice to have more details about how this was done, perhaps in a second Appendix. In particular, as the authors note elsewhere, the bar pattern varies quite a

**ESurfD**
bit along the flume. This is clear in Figure 2 and is strongest for the high discharges, where the wavelength seems to change as well. Were the bars ensemble-averaged over the whole channel length, or just the downstream part? If the whole length was used, how (if at all) were the varying bar shapes rectified relative to one another? 303 'firsts' to 'first' 304 'for calculating' to 'calculation of' or 'us to calculate' 382, 395 see 29 above 399 'from' to 'of' 416 'to' to 'with' 446 'tends' to 'tend' 448 'gives' to 'give' 491 'associate to' to 'associated with' 502ff 'Overall. . . history' This final statement is too vague to provide any useful information. The paper would be stronger if it ended with the seven clear, specific conclusions it has now. But if the authors want to end with something more general, it would be better to come up with a statement that has a memorable and useful message.

---

## Referee Comment (RC2) · Eric Prokocki (Referee) · 15 Jun 2020

Summary of Review

Eric Prokocki University of Texas at Austin ewaschle2@gmail.com

This manuscript was enjoyable to read and was quite interesting. I have no serious concerns with the current manuscript, and I am a proponent of using 'signal analyses' to examine physical phenomena, and thus I applaud the authors for using Fourier analyses to provide a relatively objective quantifiable metric, or signal, to look for when evaluating alternate bars. This technique certainly has many other applications beyond bar theory.

Overall, I recommend publication of this manuscript after 'minor' revisions (see follow-

ing).

There are a number of very 'minor' corrections, however, which need addressing. There are several structural/grammatical errors, which I have provided possible corrections for (or are highlighted) in the 'Track Changes' PDF file accompanying this general review document. In addition, I have posed a number of comments/questions (see below) that the authors may want to explore to 'round out' their current manuscript, and to connect their findings more generally to the inherent grain size continuum observed when considering the range of naturally occurring fluvial systems.

Main Focus of Manuscript

1. To compare, contrast, and test, existing alternate bar formation models and observations with newly acquired experimental alternate bar results accompanied by Fourier analyses.

Research Questions

1. How do geometrical properties of alternate bars depend on water discharge?

2. Is it possible to identify different bar styles depending on flow conditions?

3. Is there a sharp transition from alternate bar morphology to a plane-bed configuration?

General Reviewer Comments and Questions

1. The authors use the words 'bars' and 'bedforms' interchangeably throughout the manuscript. These two features are not 'identical'. In the 'Track Changes' PDF file uploaded with this general review, I have provided suggestions throughout the manuscript as to how to correct this error. In general, 'bedforms' refer to ripples and dunes (do not deflect, or steer, a large percentage of channel water mass), whilst 'bars', or 'barforms' do deflect, or steer, a large percentage of channel water mass. Thus, when the authors refer to the entire spectrum of morphologic elements of a bed (i.e., bedforms to

barforms) they should use a generic, or more neutral, term such as: 'bed features', or 'morphologic bed features', whilst when they refer to specific elements they see in their experiments they should use: i) bars or barforms, or ii) bedforms, under the appropriate conditions. I have made many of these corrections in the 'Track Changes' PDF file, but the authors should review this throughout their manuscript to ensure consistency.

2. What other morphodynamic applications can the Fourier analysis methods presented in this study be used for??

-This is an important aspect of the paper for the community at large, and it would be useful for the authors to address this question in a few lines of text.

3. When expanding the results of this study to other systems (e.g., low slope, smaller grain size), what becomes more important the Froude #, sediment transport mode (i.e., bedload, mixedload, or suspended-load), or sediment transport rates??

I realize that the answer to this question is unlikely to be straightforward, but it would be interesting to hear the author's thoughts. For instance, large deep, rivers almost always have Froude #s below criticality, and move between bedload dominated, mixed-load dominated, and suspended-load dominated, due to smaller grain size distributions and varying discharges. Are alternate bars, or diagonal bars, expected to develop under these conditions?? If so, when do these conditions arise relative to the bedload dominated experiments of this study??

-Diagonal bars are very rare, if not always absent, in silt to sand bed systems, but 'free bars' or 'alternate bars' are observed in these types of systems (especially in straight reaches)

4. Are alternate bars and diagonal bars relegated to just bedload transport conditions?? It seems that if these experiments were run at even higher flow discharges, or at steeper slopes, one would induce significant suspended-sediment transport, or high enough sediment transport rates, which potentially would erode, or 'flatten', all
'bars' leading to channel bed washout. Thus, is there a limitation on slope regarding development of alternate bars, or diagonal bars, for a given grain size??

-This limit might be better constrained by sediment transport rates as a function of shear stress??, or a higher slope just lessens the time to equilibrium conditions for a given constant discharge due to higher transport rates, or shear stresses??

-It might be interesting to think about a plot of shear stress vs grain size for given slopes to bracket, or constrain, the conditions where alternate bars, or diagonal bars, are predicted to readily form??, or not form??. Then look to see what natural systems fall within these boundaries??

5. Are the results of this study expandable as the width to depth ratio and discharge increase in step??

-To reach the identical results of this study in a flume with a larger width to depth ratio, but all other conditions equal, the discharge must be increased by some value that will equate, or scale, with the metrics measured, calculated, or observed, in this study.

-This means that there might exist a non-dimensional characteristic discharge, $Q_w$, which will recreate your results across the width to depth spectrum. For instance, this could be in the form of percentage of bankfull discharge, $Q_{bf}$ (i.e., $Q_w/Q_{bf}$)?? that will correspond to all (9) discharge conditions used in this study.

I am aware that additional experiments, or field studies, are likely necessary to resolve all, or any individual, question(s) above, but it would be useful for the author's to address this at some level in their current manuscript. I do think their results provide enough ammunition to propose a few new hypotheses, thoughts, or ways to test, the topics/ideas raised in this review.

However, I do not expect the authors to address all of the ideas or questions posed above at a deep, or significant level, they are simply 'food for thought'.

Please also note the supplement to this comment:
https://esurf.copernicus.org/preprints/esurf-2020-27/esurf-2020-27-RC2-supplement.pdf
* * *
[Figure]

**Supplement:**

[revised manuscript text omitted]

---

## Author Comment (AC1) · 13 Jul 2020

We thank the referee for the stimulating questions and the important technical suggestions.

Here we provide a response to the four main questions:

1. Apparently, there are no significant effects of the near-critical Froude number. Specifically, at the beginning of the runs small two-dimensional bedforms are sometimes observable. These can be associated with the formation of upstream-migrating antidunes, which we clearly observed in experiments with similar Shields number and relative roughness, but in a wider flume that allowed for the formation of a multi-thread system (e.g., Redolfi et al., 2017). However, they

seem to be suppressed by the formation of bars. In general, the fact that alternate bars are not directly influenced by the Froude number is a very interesting topic, which we are currently addressing (M. Redolfi, M. Musa, M. Guala, submitted to Journal of Fluid Mechanics).

2. In our experiments, flow separation seems to occur at mid-to-low discharges, due to the presence of sharp diagonal fronts. In these cases, we also observe that no sediment transport occurs downstream of the fronts.

   According to linear theories (see Colombini and Stocchino, 2012), flow separation is not a necessary ingredient for the formation of alternate and diagonal bars, which suggests that this effect probably plays a minor role at the early stages of bar formation. However, separation may be potentially important in the dynamics of fully-developed bars. For example, this effect may have an impact on the reach-averaged sediment transport, as it can induce part of the shear stress to be absorbed as form drag.

   For alternate bars, the impact of flow separation is mainly "local", thus having a minor impact on the reach-averaged properties (Colombini et al., 1987). Therefore the main effect of bars on sediment transport is rather related to the non-uniform distribution of the shear stress, which tends to significantly increase the net sediment flux (Francalanci et al., 2012).

   For diagonal bars, the effect of the form drag may be more important, and may eventually prevail over the effect of the non-uniform shear stress distribution. Our measurements seem to suggest that the transport rate we measured at the highest flow rates is indeed lower than what is expected in flat bed conditions. However, we realized that data from our experiments are not sufficient to fully support this idea, and a set of specific experimental runs would be needed.

3. We fully agree that it is interesting that transition to three dimensional dunes (i.e. diagonal bars) occurs in conditions typical of gravel-bed rivers. According to the Colombini and Stocchino (2012) theory, the key parameter controlling the transition from two- to three-dimensional dunes is actually the relative roughness ($d_{50}/D$) or equivalently the Chèzy coefficient (defined in plane-bed conditions). Specifically, their perturbation theory reveals that two-dimensional bedforms develop when the relative roughness is small, while three-dimensional oblique dunes are expected when the sediment is comparatively coarse. This is also consistent with the experiments of Jaeggi (1984), showing the formation of diagonal bars in conditions that are typical of gravel bed rivers.

In our opinion, formation of diagonal bars in rivers is discouraged because of their relatively small amplitude. In real conditions (unsteady discharge, presence of channel curvature, poorly sorted sediment), the bed morphology is expected to depend on a competition among different kinds of bedforms (diagonal bars, free bars, forced bars, dunes). In this competition, diagonal bars can be easily suppressed by the formation of other, more prominent bed features. Therefore, we may expect that diagonal bars are rather ephemeral, and observable in particular conditions only.

In the new version of the manuscript we have added a couple of sentences in the Discussion Section, to briefly introduce these important considerations.

4. We agree that an experiment encompassing a wide range of conditions would be very interesting. Specifically, the part that would deserve most attention is the transition from two- to three-dimensional bedforms. To avoid excessive Shields stress and Froude number, an optimal design would imply a variable slope, and a discharge that is adjusted in order to maintain a nearly constant shear stress, (which would also imply a relatively constant Froude number). In this case, theoretical analysis provides a very useful tool to guide the choice of the experimental conditions. Our experimental facility is not ideal for this kind of experiment, because the slope of the flume is not easily adjustable, and the banks do not allow for a lateral view of the bed profile. However, we think that in general this experiment is absolutely feasible.

In the revised manuscript we have addressed all the technical questions. Specifically:

- we have implemented all the suggested grammatical corrections, except for the British spelling (ESurf allows the author to use their preferred spelling);

- we have added Shields parameter and Rouse number to Table 1;

- we have used a more neutral term instead of "this kind of bedform";

- we have rearranged Section 2.4, to better explain what we changed with respect to the original formulation of Colombini et al. (1987).

- we have better explained the definition of ensemble bar;

- since we didn't find a memorable concluding message, we fully agree it is better to end the paper with the specific conclusion points.

**References**

Colombini, M. and Stocchino, A.: Three-dimensional river bed forms, Journal of Fluid Mechanics, 695, 63–80, 10.1017/jfm.2011.556, 2012.

Colombini, M., Seminara, G., and Tubino, M.: Finite-amplitude alternate bars, Journal of Fluid Mechanics, 181, 213, 10.1017/S0022112087002064, 1987.

Francalanci, S., Solari, L., Toffolon, M., and Parker, G.: Do alternate bars affect sediment transport and flow resistance in gravel-bed rivers?, Earth Surface Processes and Landforms, 37, 866–875, 10.1002/esp.3217, 2012.
Jaeggi, M. N. R.: Formation and Effects of Alternate Bars, Journal of Hydraulic Engineering, 110, 142–156, 10.1061/(ASCE)0733-9429(1984)110:2(142), 1984.

Redolfi, M., Guidorizzi, L., Tubino, M., and Bertoldi, W.: Capturing the spatiotemporal variability of bed load transport: a time-lapse imagery technique, Earth Surface Processes and Landforms, 10.1002/esp.4126, http://doi.wiley.com/10.1002/esp.4126, 2017.

---

## Author Comment (AC2) · 13 Jul 2020

We thank the referee for the interesting questions and for the extensive grammatical and structural corrections. In the revised manuscript, we have included nearly all the minor corrections.

Here we provide the response to the five general comments:

1. From a technical point of view, bars are actually a particular kind of bedform, as they are the product on an altimetric instability of the bed. Specifically, they are often referred as large-scale bedforms (e.g. Jaeggi, 1984; Fujita and Mu-ramoto, 1985; Church and Rice, 2009). For example, this it clear from the paper of Colombini and Stocchino (2012), titled "Three dimensional river bed forms",

which includes alternate bars, diagonal bars (i.e. 3-D oblique dunes), and 2-D dunes in a unified theoretical framework. However, we understand that the use of the word "bedforms" interchangeably may create confusion, as in most cases this term is adopted to indicate small scale-bed features. To find a compromise between being consistent with previous literature and avoiding confusion, we followed the approach proposed by the Referee #1, employing a more neutral term when possible.

2. In general the two-dimensional Fourier analysis can be used to study any spatial signal, and has been often employed in river morphodynamic studies (e.g., Repetto et al., 2002; Porcile et al., 2020). Probably the main peculiarity of our methodology is rather the definition of the "ensemble bar" as the average topography of multiple bar wavelengths, which is then analysed through the Fourier transform. In the revised manuscript, we have added a sentence in the last point of the Conclusions, to highlight the usefulness of the approach for different applications.

3. From theoretical works, it clearly appears that the key parameter controlling the formation of alternate bars is the channel width-to-depth ratio (e.g., Fredsoe, 1978; Colombini et al., 1987). This is because in relatively narrow channels the effect of the lateral bed slope on the bedload transport is proportionally more important, and it acts as a stabilizing effect that tends to flatten the bottom. Other parameters (especially the Shields number and the relative roughness) are also important, but bars are expected to form for a wide range of these parameters, provided the width-to-depth ratio is sufficiently large (see Figure 6 of Colombini et al., 1987). Specifically, there is not an upper limit of the Shields number for the formation of alternate bars. As a consequence, alternate bars are definitely expected to form in sand bed rivers (e.g., Bertagni and Camporeale, 2018), often coexisting with dunes, as also highlighted by the Referee #1.

Far less information exists on conditions for the existence of diagonal bars. How-

**ESurfD**
ever, the analysis of Colombini and Stocchino (2012) suggests that diagonal bars (i.e. 3-D oblique dunes) tend to form when the sediment is relatively coarse, while classic, 2-D dunes are expected in sand bed channels. This is also consistent with the experimental results of Jaeggi (1984), showing formation of diagonal bars in conditions that are typical of gravel bed rivers.

4. Alternate bars and diagonal bars are not relegated to just bedload transport conditions. We agree that increasing discharge in our experiments would induce significant suspended transport (see values of the Rouse number we have added to Table 1). However, as explained above, there is no reason to associate suspended load with the disappearance of bars. Moreover, at high transport rates the stabilizing effect of the transverse slope becomes weaker (see our Eq. (8)), which tends to even promote the formation of bars. Therefore, interpreting the disappearance of bars we observed at high discharge with the capacity of the flow to "flatten" the bed is not correct. In general, bar formation is crucially dependent on the width-to-depth ratio. For this reason, it is not possible to identify limits merely based on transport rate, slope, or relative roughness.

   We agree with the referee that a higher transport rate lessens the time to equilibrium conditions. This can be an important factor when studying the bar adaptation to unsteady flow conditions, but not for determining the equilibrium bar properties.

5. The width-to-depth ratio is the key controlling parameter. Therefore, it is not possible to reach identical results with a different width-to-depth ratio and other conditions equal. In more practical terms, we can say that the bars dynamics crucially depends on the channel width. Specifically, varying the channel width by keeping the other conditions (slope, water depth, Shields number) fixed would result in a very different response of bars.

   We understand that thinking in terms of discharge may help to simplify the problem. However, knowing the percentage of bankfull discharge is not sufficient,
because this parameter does not take into account the channel width. This is the reason for which we introduced the scaled discharge $\Delta Q^* = (Q - Q_{cr})/(Q_{cr} - Q_i)$. Since the critical discharge $Q_{cr}$ highly depends on the channel width, the scaled discharge $\Delta Q^*$ contains the essential information needed to measure the possibility of bars to form.

**References**

Bertagni, M. B. and Camporeale, C.: Finite amplitude of free alternate bars with suspended load, Water Resources Research, 10.1029/2018WR022819, 2018.

Church, M. and Rice, S. P.: Form and growth of bars in a wandering gravel-bed river, Earth Surface Processes and Landforms, 34, 1422–1432, 10.1002/esp.1831, 2009.

Colombini, M. and Stocchino, A.: Three-dimensional river bed forms, Journal of Fluid Mechanics, 695, 63–80, 10.1017/jfm.2011.556, 2012.

Colombini, M., Seminara, G., and Tubino, M.: Finite-amplitude alternate bars, Journal of Fluid Mechanics, 181, 213, 10.1017/S0022112087002064, 1987.

Fredsoe, J.: Meandering and Braiding of Rivers, Journal of Fluid Mechanics, 84, 609–624, 10.1017/S0022112078000373, 1978.

Fujita, Y. and Muramoto, Y.: Studies on the Process of Development of Alternate Bars, Tech. Rep. 3, 1985.

Jaeggi, M. N. R.: Formation and Effects of Alternate Bars, Journal of Hydraulic Engineering, 110, 142–156, 10.1061/(ASCE)0733-9429(1984)110:2(142), 1984.

Porcile, G., Blondeaux, P., and Colombini, M.: Starved versus alluvial river bedforms: an experimental investigation, Earth Surface Processes and Landforms, 10.1002/esp.4800, 2020.

Repetto, R., Tubino, M., and Paola, C.: Planimetric instability of channels with variable width, Journal of Fluid Mechanics, 457, 79–109, 10.1017/S0022112001007595, http://journals.cambridge.org/article{_}S0022112001007595, 2002.